# A statistical model for tensor PCA

**Andrea Montanari**
Statistics & Electrical Engineering
Stanford University

**Emile Richard**
Electrical Engineering
Stanford University

## Abstract

We consider the Principal Component Analysis problem for large tensors of arbitrary order $k$ under a single-spike (or rank-one plus noise) model. On the one hand, we use information theory, and recent results in probability theory, to establish necessary and sufficient conditions under which the principal component can be estimated using unbounded computational resources. It turns out that this is possible as soon as the signal-to-noise ratio $\beta$ becomes larger than $C\sqrt{k \log k}$ (and in particular $\beta$ can remain bounded as the problem dimensions increase).

On the other hand, we analyze several polynomial-time estimation algorithms, based on tensor unfolding, power iteration and message passing ideas from graphical models. We show that, unless the signal-to-noise ratio diverges in the system dimensions, none of these approaches succeeds. This is possibly related to a fundamental limitation of computationally tractable estimators for this problem.

We discuss various initializations for tensor power iteration, and show that a tractable initialization based on the spectrum of the unfolded tensor outperforms significantly baseline methods, statistically and computationally. Finally, we consider the case in which additional side information is available about the unknown signal. We characterize the amount of side information that allows the iterative algorithms to converge to a good estimate.

## 1 Introduction

Given a data matrix $\mathbf{X}$, Principal Component Analysis (PCA) can be regarded as a 'denoising' technique that replaces $\mathbf{X}$ by its closest rank-one approximation. This optimization problem can be solved efficiently, and its statistical properties are well-understood. The generalization of PCA to tensors is motivated by problems in which it is important to exploit higher order moments, or data elements are naturally given more than two indices. Examples include topic modeling, video processing, collaborative filtering in presence of temporal/context information, community detection [1], spectral hypergraph theory. Further, finding a rank-one approximation to a tensor is a bottleneck for tensor-valued optimization algorithms using conditional gradient type of schemes. While tensor factorization is NP-hard [11], this does not necessarily imply intractability for natural statistical models. Over the last ten years, it was repeatedly observed that either convex optimization or greedy methods yield optimal solutions to statistical problems that are intractable from a worst case perspective (well-known examples include sparse regression and low-rank matrix completion).

In order to investigate the fundamental tradeoffs between computational resources and statistical power in tensor PCA, we consider the simplest possible model where this arises, whereby an unknown unit vector $\mathbf{v_0}$ is to be inferred from noisy multilinear measurements. Namely, for each unordered $k$-uple $\{i_1, i_2, \ldots, i_k\} \subseteq [n]$, we measure

$$\mathbf{X}_{i_1,i_2,\ldots,i_k} = \beta(\mathbf{v_0})_{i_1}(\mathbf{v_0})_{i_2}\cdots(\mathbf{v_0})_{i_k} + \mathbf{Z}_{i_1,i_2,\ldots,i_k}, \tag{1}$$

with $\mathbf{Z}$ Gaussian noise (see below for a precise definition) and wish to reconstruct $\mathbf{v_0}$. In tensor notation, the observation model reads (see the end of this section for notations)

$$\mathbf{X} = \beta\, \mathbf{v_0}^{\otimes k} + \mathbf{Z} \ . \qquad\qquad \text{Spiked Tensor Model}$$

This is analogous to the so called 'spiked covariance model' used to study matrix PCA in high dimensions [12].

It is immediate to see that maximum-likelihood estimator $\mathbf{v}^{\mathrm{ML}}$ is given by a solution of the following problem

$$\begin{aligned} \text{maximize} \quad & \langle \mathbf{X}, \mathbf{v}^{\otimes k} \rangle, \\ \text{subject to} \quad & \|\mathbf{v}\|_2 = 1 \,. \end{aligned} \qquad \text{Tensor PCA}$$

Solving it exactly is –in general– NP hard [11].

We next summarize our results. Note that, given a completely observed rank-one symmetric tensor $\mathbf{v_0}^{\otimes k}$ (i.e. for $\beta = \infty$), it is easy to recover the vector $\mathbf{v_0} \in \mathbb{R}^n$. It is therefore natural to ask the question *for which signal-to-noise ratios can one still reliably estimate $\mathbf{v_0}$?* The answer appears to depend dramatically on the computational resources[1].

**Ideal estimation.** Assuming unbounded computational resources, we can solve the Tensor PCA optimization problem and hence implement the maximum likelihood estimator $\widehat{\mathbf{v}}^{\mathrm{ML}}$. We use recent results in probability theory to show that this approach is successful for $\beta \geq \mu_k = \sqrt{k \log k}(1 + o_k(1))$. In particular, above this threshold[2] we have, with high probability,

$$\|\widehat{\mathbf{v}}^{\mathrm{ML}} - \mathbf{v_0}\|_2^2 \lesssim \frac{2.01 \, \mu_k}{\beta} \,. \qquad (2)$$

We use an information-theoretic argument to show that no approach can do significantly better, namely no procedure can estimate $\mathbf{v_0}$ accurately for $\beta \leq c\sqrt{k}$ (for $c$ a universal constant).

**Tractable estimators: Unfolding.** We consider two approaches to estimate $\mathbf{v_0}$ that can be implemented in polynomial time. The first approach is based on tensor unfolding: starting from the tensor $\mathbf{X} \in \bigotimes^k \mathbb{R}^n$, we produce a matrix $\mathsf{Mat}(\mathbf{X})$ of dimensions $n^q \times n^{k-q}$. We then perform matrix PCA on $\mathsf{Mat}(\mathbf{X})$. We show that this method is successful for $\beta \gtrsim n^{(\lceil k/2 \rceil - 1)/2}$. A heuristics argument suggests that the necessary and sufficient condition for tensor unfolding to succeed is indeed $\beta \gtrsim n^{(k-2)/4}$ (which is below the rigorous bound by a factor $n^{1/4}$ for $k$ odd). We can indeed confirm this conjecture for $k$ even and under an asymmetric noise model.

**Tractable estimators: Warm-start power iteration and Approximate Message Passing.** We prove that, initializing power iteration uniformly at random, it converges very rapidly to an accurate estimate provided $\beta \gtrsim n^{(k-1)/2}$. A heuristic argument suggests that the correct necessary and sufficient threshold is given by $\beta \gtrsim n^{(k-2)/2}$. Motivated by the last observation, we consider a 'warm-start' power iteration algorithm, in which we initialize power iteration with the output of tensor unfolding. This approach appears to have the same threshold signal-to-noise ratio as simple unfolding, but significantly better accuracy above that threshold. We extend power iteration to an approximate message passing (AMP) algorithm [7, 4]. We show that the behavior of AMP is qualitatively similar to the one of naive power iteration. In particular, AMP fails for any $\beta$ bounded as $n \to \infty$.

**Side information.** Given the above computational complexity barrier, it is natural to study weaker version of the original problem. Here we assume that extra information about $\mathbf{v_0}$ is available. This can be provided by additional measurements or by approximately solving a related problem, for instance a matrix PCA problem as in [1]. We model this additional information as $\mathbf{y} = \gamma \mathbf{v_0} + \mathbf{g}$ (with $\mathbf{g}$ an independent Gaussian noise vector), and incorporate it in the initial condition of AMP algorithm. We characterize exactly the threshold value $\gamma_* = \gamma_*(\beta)$ above which AMP converges to an accurate estimator. The thresholds for various classes of algorithms are summarized below.

| Method | Required $\beta$ (rigorous) | Required $\beta$ (heuristic) |
|---|---|---|
| Tensor Unfolding | $O(n^{(\lceil k/2 \rceil - 1)/2})$ | $n^{(k-2)/4}$ |
| Tensor Power Iteration (with random init.) | $O(n^{(k-1)/2})$ | $n^{(k-2)/2}$ |
| Maximum Likelihood | 1 | – |
| Information-theory lower bound | 1 | – |

We will conclude the paper with some insights that we believe provide useful guidance for tensor factorization heuristics. We illustrate these insights through simulations.

## 1.1 Notations

Given $\mathbf{X} \in \bigotimes^k \mathbb{R}^n$ a real $k$-th order tensor, we let $\{\mathbf{X}_{i_1,\dots,i_k}\}_{i_1,\dots,i_k}$ denote its coordinates and define a map $\mathbf{X} : \mathbb{R}^n \to \mathbb{R}^n$, by letting, for $\mathbf{v} \in \mathbb{R}^n$,

$$\mathbf{X}\{\mathbf{v}\}_i = \sum_{j_1,\cdots,j_{k-1} \in [n]} \mathbf{X}_{i,j_1,\cdots,j_{k-1}} \, \mathbf{v}_{j_1} \cdots \mathbf{v}_{j_{k-1}} \,. \tag{3}$$

The outer product of two tensors is $\mathbf{X} \otimes \mathbf{Y}$, and, for $\mathbf{v} \in \mathbb{R}^n$, we define $\mathbf{v}^{\otimes k} = \mathbf{v} \otimes \cdots \otimes \mathbf{v} \in \bigotimes^k \mathbb{R}^n$ as the $k$-th outer power of $\mathbf{v}$. We define the inner product of two tensors $\mathbf{X}, \mathbf{Y} \in \bigotimes^k \mathbb{R}^n$ as

$$\langle \mathbf{X}, \mathbf{Y} \rangle = \sum_{i_1,\cdots,i_k \in [n]} \mathbf{X}_{i_1,\cdots,i_k} \mathbf{Y}_{i_1,\cdots,i_k} \,. \tag{4}$$

We define the Frobenius (Euclidean) norm of a tensor $\mathbf{X}$, by $\|\mathbf{X}\|_F = \sqrt{\langle \mathbf{X}, \mathbf{X} \rangle}$, and its operator norm by

$$\|\mathbf{X}\|_{op} \equiv \max\{\langle \mathbf{X}, \mathbf{u}_1 \otimes \cdots \otimes \mathbf{u}_k \rangle \, : \, \forall i \in [k] \,, \, \|\mathbf{u}_i\|_2 \leq 1\}. \tag{5}$$

For the special case $k = 2$, it reduces to the ordinary $\ell_2$ matrix operator norm. For $\pi \in \mathfrak{S}_k$, we will denote by $\mathbf{X}^\pi$ the tensor with permuted indices $\mathbf{X}^\pi_{i_1,\cdots,i_k} = \mathbf{X}_{\pi(i_1),\cdots,\pi(i_k)}$. We call the tensor $\mathbf{X}$ *symmetric* if, for any permutation $\pi \in \mathfrak{S}_k$, $\mathbf{X}^\pi = \mathbf{X}$. It is proved [23] that, for symmetric tensors, the value of problem Tensor PCA coincides with $\|\mathbf{X}\|_{op}$ up to a sign. More precisely, for symmetric tensors we have the equivalent representation $\max\{|\langle \mathbf{X}, \mathbf{u}^{\otimes k} \rangle| \, : \, \|\mathbf{u}\|_2 \leq 1\}$. We denote by $\mathbf{G} \in \bigotimes^k \mathbb{R}^n$ a tensor with independent and identically distributed entries $\mathbf{G}_{i_1,\cdots,i_k} \sim \mathsf{N}(0,1)$ (note that this tensor is not symmetric). We define the *symmetric standard normal* noise tensor $\mathbf{Z} \in \bigotimes^k \mathbb{R}^n$ by

$$\mathbf{Z} = \frac{1}{k!} \sqrt{\frac{k}{n}} \sum_{\pi \in \mathfrak{S}_k} \mathbf{G}^\pi \,. \tag{6}$$

We use the loss function

$$\mathsf{Loss}(\widehat{\mathbf{v}}, \mathbf{v_0}) \equiv \min \left( \|\widehat{\mathbf{v}} - \mathbf{v_0}\|_2^2, \|\widehat{\mathbf{v}} + \mathbf{v_0}\|_2^2 \right) = 2 - 2|\langle \widehat{\mathbf{v}}, \mathbf{v_0} \rangle| \,. \tag{7}$$

## 2 Ideal estimation

In this section we consider the problem of estimating $\mathbf{v_0}$ under the observation model Spiked Tensor Model, when no constraint is imposed on the complexity of the estimator. Our first result is a lower bound on the loss of *any* estimator.

**Theorem 1.** *For any estimator $\widehat{\mathbf{v}} = \widehat{\mathbf{v}}(\mathbf{X})$ of $\mathbf{v_0}$ from data $\mathbf{X}$, such that $\|\widehat{\mathbf{v}}(\mathbf{X})\|_2 = 1$ (i.e. $\widehat{\mathbf{v}} : \otimes^k \mathbb{R}^n \to \mathbb{S}^{n-1}$), we have, for all $n \geq 4$,*

$$\beta \leq \sqrt{\frac{k}{10}} \quad \Rightarrow \quad \mathbb{E} \, \mathsf{Loss}(\widehat{\mathbf{v}}, \mathbf{v_0}) \geq \frac{1}{32} \,. \tag{8}$$

In order to establish a matching upper bound on the loss, we consider the maximum likelihood estimator $\widehat{\mathbf{v}}^{\mathrm{ML}}$, obtained by solving the Tensor PCA problem. As in the case of matrix denoising, we expect the properties of this estimator to depend on signal to noise ratio $\beta$, and on the 'norm' of the noise $\|\mathbf{Z}\|_{op}$ (i.e. on the value of the optimization problem Tensor PCA in the case $\beta = 0$). For the

matrix case $k = 2$, this coincides with the largest eigenvalue of $\mathbf{Z}$. Classical random matrix theory shows that –in this case– $\|\mathbf{Z}\|_{op}$ concentrates tightly around 2 [10, 6, 3].

It turns out that tight results for $k \geq 3$ follow immediately from a technically sophisticated analysis of the stationary points of random Morse functions by Auffinger, Ben Arous and Cerny [2].

**Lemma 2.1.** *There exists a sequence of real numbers $\{\mu_k\}_{k \geq 2}$, such that*

$$\limsup_{n \to \infty} \|\mathbf{Z}\|_{op} \leq \mu_k \quad (k \text{ odd}), \tag{9}$$

$$\lim_{n \to \infty} \|\mathbf{Z}\|_{op} = \mu_k \quad (k \text{ even}). \tag{10}$$

*Further $\|\mathbf{Z}\|_{op}$ concentrates tightly around its expectation. Namely, for any $n, k$*

$$\mathbb{P}\big(\big|\|\mathbf{Z}\|_{op} - \mathbb{E}\|\mathbf{Z}\|_{op}\big| \geq s\big) \leq 2\,e^{-ns^2/(2k)} \,. \tag{11}$$

*Finally $\mu_k = \sqrt{k \log k}(1 + o_k(1))$ for large $k$.*

For instance, a large order-3 Gaussian tensor should have $\|\mathbf{Z}\|_{op} \approx 2.87$, while a large order 10 tensor has $\|\mathbf{Z}\|_{op} \approx 6.75$. As a simple consequence of Lemma 2.1, we establish an upper bound on the error incurred by the maximum likelihood estimator.

**Theorem 2.** *Let $\mu_k$ be the sequence of real numbers introduced above. Letting $\widehat{\mathbf{v}}^{\mathrm{ML}}$ denote the maximum likelihood estimator (i.e. the solution of Tensor PCA), we have for $n$ large enough, and all $s > 0$*

$$\beta \geq \mu_k \Rightarrow \mathsf{Loss}(\widehat{\mathbf{v}}^{\mathrm{ML}}, \mathbf{v_0}) \leq \frac{2}{\beta}(\mu_k + s)\,, \tag{12}$$

*with probability at least $1 - 2e^{-ns^2/(16k)}$.*

The following upper bound on the value of the Tensor PCA problem is proved using Sudakov-Fernique inequality. While it is looser than Lemma 2.1 (corresponding to the case $\beta = 0$), we expect it to become sharp for $\beta \geq \beta_k$ a suitably large constant.

**Lemma 2.2.** *Under Spiked Tensor Model model, we have*

$$\limsup_{n \to \infty} \mathbb{E}\|\mathbf{Z}\|_{op} \leq \max_{\tau \geq 0}\left\{\beta\left(\frac{\tau}{\sqrt{1 + \tau^2}}\right)^k + \frac{k}{\sqrt{1 + \tau^2}}\right\}\,. \tag{13}$$

*Further, for any $s \geq 0$,*

$$\mathbb{P}\big(\big|\|\mathbf{Z}\|_{op} - \mathbb{E}\|\mathbf{Z}\|_{op}\big| \geq s\big) \leq 2\,e^{-ns^2/(2k)} \,. \tag{14}$$

## 3 Tensor Unfolding

A simple and popular heuristics to obtain tractable estimators of $\mathbf{v_0}$ consists in constructing a suitable matrix with the entries of $\mathbf{X}$, and performing PCA on this matrix.

### 3.1 Symmetric noise

For an integer $k \geq q \geq k/2$, we introduce the unfolding (also referred to as matricization or reshape) operator $\mathsf{Mat}_q : \otimes^k \mathbb{R}^n \to \mathbb{R}^{n^q \times n^{k-q}}$ as follows. For any indices $i_1, i_2, \cdots, i_k \in [n]$, we let $a = 1 + \sum_{j=1}^{q}(i_j - 1)n^{j-1}$ and $b = 1 + \sum_{j=q+1}^{k}(i_j - 1)n^{j-q-1}$, and define

$$[\mathsf{Mat}_q(\mathbf{X})]_{a,b} = \mathbf{X}_{i_1, i_2, \cdots, i_k} \,. \tag{15}$$

Standard convex relaxations of low-rank tensor estimation problem compute factorizations of $\mathsf{Mat}_q(\mathbf{X})$[22, 15, 17, 19]. Not all unfoldings (choices of $q$) are equivalent. It is natural to expect that this approach will be successful only if the signal-to-noise ratio exceeds the operator norm of the unfolded noise $\|\mathsf{Mat}_q(\mathbf{Z})\|_{op}$. The next lemma suggests that the latter is minimal when $\mathsf{Mat}_q(\mathbf{Z})$ is 'as square as possible' . A similar phenomenon was observed in a different context in [17].

**Lemma 3.1.** *For any integer $k/2 \leq q \leq k$ we have, for some universal constant $C_k$,*

$$\frac{1}{\sqrt{(k-1)!}}\, n^{(q-1)/2} \left(1 - \frac{C_k}{n^{\max(q,k-q))}}\right) \leq \mathbb{E}\|\mathsf{Mat}_q(\mathbf{Z})\|_{op} \leq \sqrt{k}\left(n^{(q-1)/2} + n^{(k-q-1)/2}\right) \ . \tag{16}$$

*For all $n$ large enough, both bounds are minimized for $q = \lceil k/2 \rceil$. Further*

$$\mathbb{P}\Big\{\big|\|\mathsf{Mat}_q(\mathbf{Z})\|_{op} - \mathbb{E}\|\mathsf{Mat}_q(\mathbf{Z})\|_{op}\big| \geq t\Big\} \leq 2\, e^{-nt^2/(2k)}\,. \tag{17}$$

The last lemma suggests the choice $q = \lceil k/2 \rceil$, which we shall adopt in the following, unless stated otherwise. We will drop the subscript from $\mathsf{Mat}$.

Let us recall the following standard result derived directly from Wedin perturbation Theorem [24], and stated in the context of the spiked model.

**Theorem 3** (Wedin perturbation). *Let $\mathbf{M} = \beta\mathbf{u_0}\mathbf{w_0}^\mathsf{T} + \boldsymbol{\Xi} \in \mathbb{R}^{m \times p}$ be a matrix with $\|\mathbf{u_0}\|_2 = \|\mathbf{w_0}\|_2 = 1$. Let $\widehat{\mathbf{w}}$ denote the right singular vector of $\mathbf{M}$. If $\beta > 2\|\boldsymbol{\Xi}\|_{op}$, then*

$$\mathsf{Loss}(\widehat{\mathbf{w}}, \mathbf{w_0}) \leq \frac{8\|\boldsymbol{\Xi}\|_{op}^2}{\beta^2} \ . \tag{18}$$

**Theorem 4.** *Letting $\mathbf{w} = \mathbf{w}(\mathbf{X})$ denote the top right singular vector of $\mathsf{Mat}(\mathbf{X})$, we have the following, for some universal constant $C = C_k > 0$, and $b \equiv (1/2)(\lceil k/2 \rceil - 1)$.*

*If $\beta \geq 5\, k^{1/2}\, n^b$ then, with probability at least $1 - n^{-2}$, we have*

$$\mathsf{Loss}\Big(\mathbf{w}, \mathsf{vec}\big(\mathbf{v_0}^{\otimes\lfloor k/2 \rfloor}\big)\Big) \leq \frac{C\, kn^{2b}}{\beta^2}\,. \tag{19}$$

### 3.2 Asymmetric noise and recursive unfolding

A technical complication in analyzing the random matrix $\mathsf{Mat}_q(\mathbf{X})$ lies in the fact that its entries are not independent, because the noise tensor $\mathbf{Z}$ is assumed to be symmetric. In the next theorem we consider the case of non-symmetric noise and even $k$. This allows us to leverage upon known results in random matrix theory [18, 8, 5] to obtain: $(i)$ Asymptotically sharp estimates on the critical signal-to-noise ratio; $(ii)$ A lower bound on the loss below the critical signal-to-noise ratio. Namely, we consider observations

$$\widetilde{\mathbf{X}} = \beta\mathbf{v_0}^{\otimes k} + \frac{1}{\sqrt{n}}\mathbf{G}\,. \tag{20}$$

where $\mathbf{G} \in \otimes^k \mathbb{R}^n$ is a standard Gaussian tensor (i.e. a tensor with i.i.d. standard normal entries).

Let $\mathbf{w} = \mathbf{w}(\widetilde{\mathbf{X}}) \in \mathbb{R}^{n^{k/2}}$ denote the top right singular vector of $\mathsf{Mat}(\mathbf{X})$. For $k \geq 4$ even, and define $b \equiv (k-2)/4$, as above. By [18, Theorem 4], or [5, Theorem 2.3], we have the following almost sure limits

$$\beta \leq (1 - \varepsilon)n^b \Rightarrow \quad \lim_{n \to \infty}\langle\mathbf{w}(\widetilde{\mathbf{X}}), \mathsf{vec}(\mathbf{v_0}^{\otimes(k/2)})\rangle = 0\,, \tag{21}$$

$$\beta \geq (1 + \varepsilon)n^b \Rightarrow \quad \liminf_{n \to \infty}\big|\langle\mathbf{w}(\widetilde{\mathbf{X}}), \mathsf{vec}(\mathbf{v_0}^{\otimes(k/2)})\rangle\big| \geq \sqrt{\frac{\varepsilon}{1 + \varepsilon}}\,. \tag{22}$$

In other words $\mathbf{w}(\widetilde{\mathbf{X}})$ is a good estimate of $\mathbf{v_0}^{\otimes(k/2)}$ if and only if $\beta$ is larger than $n^b$.

We can use $\mathbf{w}(\widetilde{\mathbf{X}}) \in \mathbb{R}^{2b+1}$ to estimate $\mathbf{v_0}$ as follows. Construct the unfolding $\mathsf{Mat}_1(\mathbf{w}) \in \mathbb{R}^{n \times n^{2b}}$ (slight abuse of notation) of $\mathbf{w}$ by letting, for $i \in [n]$, and $j \in [n^{2b}]$,

$$\mathsf{Mat}_1(\mathbf{w})_{i,j} = \mathbf{w}_{i+(j-1)n}\,, \tag{23}$$

we then let $\widehat{\mathbf{v}}$ to be the left principal vector of $\mathsf{Mat}_1(\mathbf{X})$. We refer to this algorithm as to *recursive unfolding*.

**Theorem 5.** *Let $\widetilde{\mathbf{X}}$ be distributed according to the non-symmetric model (20) with $k \geq 4$ even, define $b \equiv (k-2)/4$. and let $\widehat{\mathbf{v}}$ be the estimate obtained by two-steps recursive matricization.*

*If $\beta \geq (1+\varepsilon)n^b$ then, almost surely*

$$\lim_{n \to \infty} \mathsf{Loss}(\widehat{\mathbf{v}}, \mathbf{v_0}) = 0 \,. \tag{24}$$

We conjecture that the weaker condition $\beta \gtrsim n^{(k-2)/4}$ is indeed sufficient also for our original symmetric noise model, both for $k$ even and for $k$ odd.

## 4 Power Iteration

Iterating over (multi-) linear maps induced by a (tensor) matrix is a standard method for finding leading eigenpairs, see [14] and references therein for tensor-related results. In this section we will consider a simple power iteration, and then its possible uses in conjunction with tensor unfolding. Finally, we will compare our analysis with results available in the literature.

### 4.1 Naive power iteration

The simplest iterative approach is defined by the following recursion

$$\mathbf{v}^0 = \frac{\mathbf{y}}{\|\mathbf{y}\|_2} \,, \quad \text{and} \quad \mathbf{v}^{t+1} = \frac{\mathbf{X}\{\mathbf{v}^t\}}{\|\mathbf{X}\{\mathbf{v}^t\}\|_2} \,. \qquad\qquad \text{Power Iteration}$$

The following result establishes convergence criteria for this iteration, first for generic noise $\mathbf{Z}$ and then for standard normal noise (using Lemma 2.1).

**Theorem 6.** *Assume*

$$\beta \geq 2\,e(k-1)\,\|\mathbf{Z}\|_{op}\,, \tag{25}$$

$$\frac{\langle \mathbf{y}, \mathbf{v_0} \rangle}{\|\mathbf{y}\|_2} \geq \left[ \frac{(k-1)\|\mathbf{Z}\|_{op}}{\beta} \right]^{1/(k-1)} \,. \tag{26}$$

*Then for all $t \geq t_0(k)$, the power iteration estimator satisfies $\mathsf{Loss}(\mathbf{v}^t, \mathbf{v_0}) \leq 2e\|\mathbf{Z}\|_{op}/\beta$. If $\mathbf{Z}$ is a standard normal noise tensor, then conditions (25), (26) are satisfied with high probability provided*

$$\beta \geq 2ek\,\mu_k = 6\sqrt{k^3 \log k}\,\left(1 + o_k(1)\right)\,, \tag{27}$$

$$\frac{\langle \mathbf{y}, \mathbf{v_0} \rangle}{\|\mathbf{y}\|_2} \geq \left[ \frac{k\mu_k}{\beta} \right]^{1/(k-1)} = \beta^{-1/(k-1)}\left(1 + o_k(1)\right)\,. \tag{28}$$

In Section 6 we discuss two aspects of this result: $(i)$ The requirement of a positive correlation between initialization and ground truth ; $(ii)$ Possible scenarios under which the assumptions of Theorem 6 are satisfied.

## 5 Asymptotics via Approximate Message Passing

Approximate message passing (AMP) algorithms [7, 4] proved successful in several high-dimensional estimation problems including compressed sensing, low rank matrix reconstruction, and phase retrieval [9, 13, 20, 21]. An appealing feature of this class of algorithms is that their high-dimensional limit can be characterized exactly through a technique known as 'state evolution.' Here we develop an AMP algorithm for tensor data, and its state evolution analysis focusing on the fixed $\beta$, $n \to \infty$ limit. Proofs follows the approach of [4] and will be presented in a journal publication.

In a nutshell, our AMP for Tensor PCA can be viewed as a sophisticated version of the power iteration method of the last section. With the notation $f(\mathbf{x}) = \mathbf{x}/\|\mathbf{x}\|_2$, we define the AMP iteration over vectors $\mathbf{v}^t \in \mathbb{R}^n$ by $\mathbf{v}^0 = \mathbf{y}$, $f(\mathbf{v}^{-1}) = 0$, and

$$\begin{cases} \mathbf{v}^{t+1} = \mathbf{X}\{f(\mathbf{v}^t)\} - \mathsf{b}_t\,f(\mathbf{v}^{t-1})\,, \\ \mathsf{b}_t = (k-1)\left(\langle f(\mathbf{v}^t), f(\mathbf{v}^{t-1})\rangle\right)^{k-2}\,. \end{cases} \qquad\qquad \text{AMP}$$

Our main conclusion is that *the behavior of AMP is qualitatively similar to the one of power iteration.* However, we can establish stronger results in two respects:

1. We can prove that, unless side information is provided about the signal $\mathbf{v_0}$, the AMP estimates remain essentially orthogonal to $\mathbf{v_0}$, for any fixed number of iterations. This corresponds to a converse to Theorem 6.

2. Since state evolution is asymptotically exact, we can prove sharp phase transition results with explicit characterization of their locations.

We assume that the additional information takes the form of a noisy observation $\mathbf{y} = \gamma \mathbf{v_0} + \mathbf{z}$, where $\mathbf{z} \sim \mathsf{N}(0, \mathsf{I}_n/n)$. Our next results summarize the state evolution analysis.

**Proposition 5.1.** *Let $k \geq 2$ be a fixed integer. Let $\{\mathbf{v_0}(n)\}_{n \geq 1}$ be a sequence of unit norm vectors $\mathbf{v_0}(n) \in \mathbb{S}^{n-1}$. Let also $\{\mathbf{X}(n)\}_{n \geq 1}$ denote a sequence of tensors $\mathbf{X}(n) \in \otimes^k \mathbb{R}^n$ generated following Spiked Tensor Model. Finally, let $\mathbf{v}^t$ denote the $t$-th iterate produced by AMP, and consider its orthogonal decomposition*

$$\mathbf{v}^t = \mathbf{v}^t_\| + \mathbf{v}^t_\perp \,, \tag{29}$$

*where $\mathbf{v}^t_\|$ is proportional to $\mathbf{v_0}$, and $\mathbf{v}^t_\perp$ is perpendicular. Then $\mathbf{v}^t_\perp$ is uniformly random, conditional on its norm. Further, almost surely*

$$\lim_{n \to \infty} \langle \mathbf{v}^t, \mathbf{v_0} \rangle = \lim_{n \to \infty} \langle \mathbf{v}^t_\|, \mathbf{v_0} \rangle = \tau_t \,, \tag{30}$$

$$\lim_{n \to \infty} \|\mathbf{v}^t_\perp\|_2 = 1 \,, \tag{31}$$

*where $\tau_t$ is given recursively by letting $\tau_0 = \gamma$ and, for $t \geq 0$ (we refer to this as to* state evolution*):*

$$\tau_{t+1}^2 = \beta^2 \left( \frac{\tau_t^2}{1 + \tau_t^2} \right)^{k-1} \,. \tag{32}$$

The following result characterizes the minimum required additional information $\gamma$ to allow AMP to escape from those undesired local optima. We will say that $\{\mathbf{v}^t\}_t$ converges almost surely to a *desired local optimum* if,

$$\lim_{t \to \infty} \lim_{n \to \infty} \mathsf{Loss}(\mathbf{v}^t/\|\mathbf{v}^t\|_2, \mathbf{v_0}) \leq \frac{4}{\beta^2} \,.$$

**Theorem 7.** *Consider the Tensor PCA problem with $k \geq 3$ and*

$$\beta > \omega_k \equiv \sqrt{(k-1)^{k-1}/(k-2)^{k-2}} \sim \sqrt{ek} \,.$$

*Then AMP converges almost surely to a desired local optimum if and only if $\gamma > \sqrt{1/\epsilon_k(\beta) - 1}$ where $\epsilon_k(\beta)$ is the largest solution of $(1 - \epsilon)^{(k-2)}\epsilon = \beta^{-2}$,*

In the special case $k = 3$, and $\beta > 2$, assuming $\gamma > \beta(1/2 - \sqrt{1/4 - 1/\beta^2})$, AMP tends to a desired local optimum. Numerically $\beta > 2.69$ is enough for AMP to achieve $\langle \mathbf{v_0}, \widehat{\mathbf{v}} \rangle \geq 0.9$ if $\gamma > 0.45$.

As a final remark, we note that the methods of [16] can be used to show that, under the assumptions of Theorem 7, for $\beta > \beta_k$ a sufficiently large constant, AMP asymptotically solves the optimization problem Tensor PCA. Formally, we have, almost surely,

$$\lim_{t \to \infty} \lim_{n \to \infty} \left| \langle \mathbf{X}, (\mathbf{v}^t)^{\otimes k} \rangle - \|\mathbf{X}\|_{op} \right| = 0. \tag{33}$$

## 6   Numerical experiments

### 6.1   Comparison of different algorithms

Our empirical results are reported in the appendix. The main findings are consistent with the theory developed above:

- Tensor power iteration (with random initialization) performs poorly with respect to other approaches that use some form of tensor unfolding. The gap widens as the dimension $n$ increases.

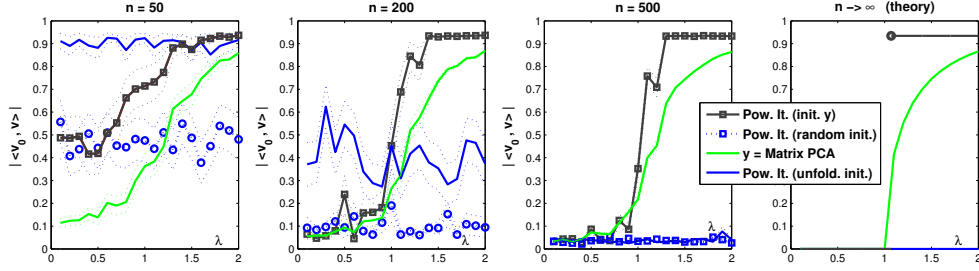

Figure 1: Simultaneous PCA at $\beta = 3$. Absolute correlation of the estimated principal component with the truth $|\langle \widehat{\mathbf{v}}, \mathbf{v_0} \rangle|$, simultaneous PCA (black) compared with matrix (green) and tensor PCA (blue).

- All algorithms based on initial unfolding (comprising PSD-constrained PCA and recursive unfolding) have essentially the same threshold. Above that threshold, those that process the singular vector (either by recursive unfolding or by tensor power iteration) have superior performances over simpler one-step algorithms.

Our heuristic arguments suggest that tensor power iteration with random initialization will work for $\beta \gtrsim n^{1/2}$, while unfolding only requires $\beta \gtrsim n^{1/4}$ (our theorems guarantee this for, respectively, $\beta \gtrsim n$ and $\beta \gtrsim n^{1/2}$). We plot the average correlation $|\langle \widehat{\mathbf{v}}, \mathbf{v_0} \rangle|$ versus (respectively) $\beta/n^{1/2}$ and $\beta/n^{1/4}$. The curve superposition confirms that our prediction captures the correct behavior already for $n$ of the order of 50.

## 6.2 The value of side information

Our next experiment concerns a simultaneous matrix and tensor PCA task: we are given a tensor $\mathbf{X} \in \otimes^3 \mathbb{R}^n$ of Spiked Tensor Model with $k = 3$ and the signal to noise ratio $\beta = 3$ is fixed. In addition, we observe $\mathbf{M} = \lambda \mathbf{v_0} \mathbf{v_0}^\mathsf{T} + \mathbf{N}$ where $\mathbf{N} \in \mathbb{R}^{n \times n}$ is a symmetric noise matrix with upper diagonal elements $i < j$ iid $\mathbf{N}_{i,j} \sim \mathsf{N}(0, 1/n)$ and the value of $\lambda \in [0, 2]$ varies. This experiment mimics a rank-1 version of topic modeling method presented in [1] where $\mathbf{M}$ is a matrix representing pairwise co-occurences and $\mathbf{X}$ triples.

The analysis in previous sections suggests to use the leading eigenvector of $\mathbf{M}$ as the initial point of AMP algorithm for tensor PCA on $\mathbf{X}$. We performed the experiments on 100 randomly generated instances with $n = 50, 200, 500$ and report in Figure 1 the mean values of $|\langle \mathbf{v_0}, \widehat{\mathbf{v}}(\mathbf{X}) \rangle|$ with confidence intervals.

Random matrix theory predicts $\lim_{n \to \infty} \langle \widehat{\mathbf{v}}_1(M), \mathbf{v_0} \rangle = \sqrt{1 - \lambda^{-2}}$ [8]. Thus we can set $\gamma = \sqrt{1 - \lambda^{-2}}$ and apply the theory of the previous section. In particular, Proposition 5.1 implies

$$\lim_{n \to \infty} \langle \widehat{\mathbf{v}}(\mathbf{X}), \mathbf{v_0} \rangle = \beta \left( 1/2 + \sqrt{1/4 - 1/\beta^2} \right) \quad \text{if} \ \gamma > \beta \left( 1/2 - \sqrt{1/4 - 1/\beta^2} \right)$$

and $\lim_{n \to \infty} \langle \widehat{\mathbf{v}}(\mathbf{X}), \mathbf{v_0} \rangle = 0$ otherwise Simultaneous PCA appears vastly superior to simple PCA. Our theory captures this difference quantitatively already for $n = 500$.

## Acknowledgements

This work was partially supported by the NSF grant CCF-1319979 and the grants AFOSR/DARPA FA9550-12-1-0411 and FA9550-13-1-0036.

## Footnotes

[1]Here we write $F(n) \lesssim G(n)$ if there exists a constant $c$ independent of $n$ (but possibly dependent on $n$, such that $F(n) \leq c \, G(n)$

[2]Note that, for $k$ even, $\mathbf{v_0}$ can only be recovered modulo sign. For the sake of simplicity, we assume here that this ambiguity is correctly resolved.

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
