[Supplementary Material · tsr_apx1.pdf]

# A statistical model for tensor PCA (Appendix)

**Andrea Montanari**
Statistics & Electrical Engineering
Stanford University

**Emile Richard**
Electrical Engineering
Stanford University

## 1   Numerical experiments

Let us emphasize two practical suggestions that arise from our work:

- Tensor unfolding is superior to tensor power iteration under our spiked model. For instance, for $k = 3$, we expect tensor power iteration to require $\beta \gtrsim n^{1/4}$ and unfolding to require $\beta \gtrsim n^{1/2}$.

- For smaller values of $\beta$, iterative methods (tensor power iteration or approximate message passing) only produce a good estimate if the initialization has a scalar product with the ground truth $\mathbf{v_0}$ that is bounded away from zero.

- As a consequence of the above, side information about the unknown vector $\mathbf{v_0}$ can greatly improve performances.

  A special case, we will study the behavior of warm start algorithms that first perform a singular value decomposition of $\mathsf{Mat}(\mathbf{X})$, and then apply an iterative method (tensor power iteration or approximate message passing).

In this section we will illustrate these suggestions through numerical simulations.

Section 1.1 describes a refinement of tensor unfolding that provides a tighter relaxation. Section 1.2 compares different algorithms.

### 1.1   PSD-constrained principal component

Note that, for $\mathbf{v} \in \mathbb{R}^n$, the outer product $\mathbf{v} \otimes \mathbf{v}$ (regarded as an $n \times n$ matrix) is positive semi-definite (PSD). Considering the case $k = 3$, we have

$$\mathsf{Mat}(\mathbf{X}) = \beta \mathsf{vec}(\mathbf{v_0} \otimes \mathbf{v_0})\, \mathbf{v_0}^{\mathsf{T}} + \mathsf{Mat}(\mathbf{Z})\,. \tag{1}$$

This remark suggests to perform a cone-constrained principal component analysis of $\mathsf{Mat}(\mathbf{X})$, where the left singular vector (viewed as a matrix) belongs to the PDS cone. In order to write this formally, it is convenient to introduce the operator $\mathtt{reshape}_{n \times n} : \mathbb{R}^{n^2} \to \mathbb{R}^{n \times n}$ that matricizes vectors as $\mathtt{reshape}_{n \times n}(\mathbf{w})_{i,j} = \mathbf{w}_{n(i-1)+j}$. The PSD-cone-constrained principal component of $\mathsf{Mat}(\mathbf{X})$, is defined by

$$(\widehat{\mathbf{w}}, \widehat{\mathbf{v}}) \doteq \arg\max \left\{ \langle \mathbf{w}, \mathsf{Mat}(\mathbf{X})\mathbf{v} \rangle \ : \ \mathtt{reshape}_{n \times n}(\mathbf{w}) \succeq 0\,, \ \|\mathbf{w}\|_2 \leq 1\,, \ \|\mathbf{v}\|_2 \leq 1 \right\} \ . \tag{2}$$

This optimization problem is NP hard, since it includes copositive programming as a special case. However [1] provides rigorous and empirical evidence that problems of this type can be solved efficiently by a projected power iteration, under statistical model on $\mathbf{X}$.

Denoting $\mathsf{P}_{\succeq} : \mathbb{R}^{n^2} \to \mathbb{R}^{n^2}$ the orthogonal projector onto the PSD cone, we iterate the following for $t \geq 0$, using random initialization of $\mathbf{u}^0 \in \mathbb{R}^n$,

$$\begin{cases} \mathbf{w}^t = \mathsf{P}_{\succeq}(\mathsf{Mat}(\mathbf{X})\mathbf{v}^t), \\ \mathbf{v}^{t+1} = \mathsf{Mat}(\mathbf{X})^{\mathsf{T}}\mathbf{w}^t / \|\mathsf{Mat}(\mathbf{X})^{\mathsf{T}}\mathbf{w}^t\|_2 \,. \end{cases} \tag{3}$$

Figure 1: Comparison of various algorithms for tensor PCA, for order 3 tensors ($k = 3$). Various curves correspond to different algorithms: unfold (simple unfolding); rec. unfold (recursive unfolding); PSD s.v. (PSD-constrained PCA); T-random (tensor power iteration with random initialization); T-rec.unf., T-PSD s.v.; T-unfold (tensor power iteration with each of the initializations above). Light dotted lines are confidence bands.

## 1.2 Comparison of different algorithms

In Fig. 1 we compare different algorithms on data generated following Spiked Tensor Model with $k = 3$, and $n \in \{25, 50, 100, 200, 400, 800\}$ and for a range of values of $\beta \in [2, 10]$. The plots represent measured values of the absolute correlation $|\langle \widehat{\mathbf{v}}, \mathbf{v_0} \rangle|$ versus $\beta$, averaged over 50 samples (except for $n = 800$, where we used 8 samples).

Figure 2: Scaling with $n$ of the threshold signal-to-noise ratio for different classes of algorithms. Left: tensor power iteration with random initialization. Right: tensor unfolding.

The main findings are consistent with the theory developed above:

- Tensor power iteration (with random initialization) performs poorly with respect to other approaches that use some form of tensor unfolding. The gap widens as the dimension $n$ increases.

- PSD-constrained principal component analysis (described in the last section) is slightly superior to plain unfolding.

- All algorithms based on initial unfolding have essentially the same threshold. Above that threshold, those that process the singular component (either by recursive unfolding or by tensor power iteration) have superior performances over simpler one-step algorithms.

In addition, we noted that the two iterative algorithms (Tensor Power Iteration and AMP) show very close behaviors in our experiments.

In Figure 2 we compare the scaling with $n$ of the threshold signal-to-noise ratio for different type of algorithms. Our heuristic arguments suggest that tensor power iteration with random initialization will work for $\beta \gtrsim n^{1/2}$, while unfolding only requires $\beta \gtrsim n^{1/4}$ (our theorems guarantee this for, respectively, $\beta \gtrsim n$ and $\beta \gtrsim n^{1/2}$). We plot the average correlation $|\langle \widehat{\mathbf{v}}, \mathbf{v_0} \rangle|$ versus (respectively) $\beta/n^{1/2}$ and $\beta/n^{1/4}$. The curve superposition confirms that our prediction captures the correct behavior already for $n$ of the order of $50$.

## Acknowledgements

This work was partially supported by the NSF grant CCF-1319979 and the grants AFOSR/DARPA FA9550-12-1-0411 and FA9550-13-1-0036.

## References

[1] Y. Deshpande, A. Montanari, and E. Richard. Cone-constrained principal component analysis. In *Neural Information Processing Systems (NIPS)*, 2014.