[Reviews · NeurIPS 2014]

Submitted by Assigned_Reviewer_22

Summary:

This paper studies the Principal Component Analysis (PCA) for large tensors of arbitrary order k under a single-spike model. Solving tensor PCA exactly is in general NP hard. Given a completely observed rank-one symmetric tensor, this paper provides conditions under which one can reliably estimate the unknown unit vector. Specifically, the paper gives conditions on signal-to-noise ratio under several scenarios which allow one to estimate the solution reliably.

For the maximum-likelihood estimator (MLE), the authors show that in an ideal case with unbounded computational resources, the MLE is successful with high probability if the signal-to-noise ration is above "sqrt(k.log(k))(1+o(1))" and for s2n ratio smaller than "sqrt(k)", no procedure can estimate the solution accurately. Moreover, the paper investigates two approaches that can be implemented in polynomial time, namely, the tensor unfolding and an approximate message passing (AMP). For the tensor unfolding method, the bound above which this method becomes effective is derived. Interestingly, the AMP fails for any s2n ratio bounded as the sample size goes to infinity. Based on the aforementioned result, the authors suggest a weaker version of the original problem by assuming that extra information about the solution is available. Given such assumption, the authors show that there exists an upper bound on the s2n ratio above which the AMP converges to the maximum-likelihood estimator.

In summary, although solving tensor PCA is in general NP hard, this paper provides a fundamental understanding of tensor PCA under a single-spike model and gives several theoretical guarantees in term of the bounds on the signal-to-noise ratio above which one can reliably estimate the solution.

Quality:

The paper is of high technical quality. All claims are equipped with sufficient proofs to verify the results presented in the paper.

Clarity:

In general, I was able to follow the logic of this paper. However, there are many obvious typos in the paper that should have been corrected.

Originality:

Although similar problems have been studied previously, the paper has additional theoretical results which improve upon the previous works.

Significance:

As tensor-based algorithms have become increasingly popular in machine learning community, I believe this paper will be useful for the following works in tensor-based algorithms.
Summary: The paper presents interesting theoretical results for tensor PCA which lies a foundation for future works along this direction. Accept.

Submitted by Assigned_Reviewer_43

Description:

The paper presents a theoretical analysis of exact and approximate ML
estimation of a simple form of tensor factorization. Namely, we are
given a tensor X (of rank k and dimension n) which equals the k-fold
tensor product of some vector v_0 with itself, plus a noise tensor
with components being iid Gaussian. The task is to estimate v_0 from
X. The task leads to a non-convex optimization problem over
v_0.

Assuming non-zero noise, the task is NP hard. The paper assumes
several known algorithms to solve the task approximately:
matricization (reshaping X to a matrix and solving matrix
factorization), power iterations, message passing, and message passing
initialized by a guiding information which is the true value of v_0
plus noise.

The following is proved:

1) The difference between the exact ML estimate (requiring exponential
time) of v_0 and the true v_0 is small with high probability if SNR is
greater than a threshold, which is a polynomial function of k. This
means, the exact ML estimation is succesfull for a reasonable SNR that
scales well with tensor rank.

2) All approximate ML estimates with no guiding information (i.e.,
matricization, power iterations, and unguided message passing) are
sucessfull only for SNR larger than an exponential function of k. That
is, for larger k these methods can yield very bad estimates.

3) The guiding information can make approximate message passing
sucessfull, provided that it the SNR of this guiding information is
reasonably low.

These results are demonstrated on simple experiments on synthetic data
and k=3.

Comments:

The achieved results are valuable and non-trivial and I recommend to
accept the paper. However, I do have a number of objections.

My most important objection: The paper presents a *statistical* analysis of exact and approximate ML estimation of tensor factorization. However, it would be helpful to
first summarize known results on hardness of the ML estimation problem
from computational complexity point of view, without any
statistics. How hard is to *approximate* the Tensor PCA
problem (on lines 59-60)? You only say that it is NP-hard, but is it in APX, does it have a PTAS? How do these results relate to your statistical analysis?

The authors informally conjecture (see abstract) that the found
dichotomy between the exact (intractable) and approximate (tractable)
ML estimation can be valid in general: all tractable algorithms for ML
estimation are inevitably unsuccesfull. It would be nice to see more
arguments for this conjecture. E.g., do you believe that all existing tractable approximate algorithms fall more or less to one of these classes?

In Theorem 3, you assume that the result of the matricization
algorithm is the top singular vector w. But this is not really the
desired result because w has yet to be (approximately) decomposed to
the (\lceil k/2\rceil)-fold tensor product of vector v_0 with
itself. Can this affect your result?

In experiments, you may wish to compute also the exact ML estimate,
which I believe is tractable for fixed k=3. This would empirically
support the results from Section 2.

In experiments, considering only k=3 may hide some important insights (how does the accuracy scale with k?).

Clarity of the paper should be improved. The authors do not help the reader to follow this mathematically very difficult paper. Some parts were apparently
written in haste. There is a number of typos, e.g.:

Theorem 1: C_1 in (7) should be C_2

Lemma 2: Model model

Reference [18] is incomplete.

Many English typos in Section 5 and in the supplement.

Finally, let me admit that the topic is not exactly my expertise and
therefore verifying the proofs in detail is beyond my abilities. The
main paper is only a summary of results and all the proofs are in the
supplement. This material is mathematically very heavy and I believe
only real specialists can understand it in detail.
Summary: The results are valuable and non-trivial. However, clarity should be improved.

Submitted by Assigned_Reviewer_44

SUMMARY: This paper studies the statistical properties of Tensor PCA. It considers a simplified model called Spiked Tensor Model, which is a rank-1 tensor (defined by a scalar beta and a unit vector v0) contaminated with Gaussian noise. The paper analyzes the effect of signal to noise ratio on reconstruction of the vector v0.
The paper provides a negative result that shows that for weak signals ( beta < sqrt{k} ), any estimator has a large loss with a high probability, i.e., cannot recover v0. It then shows that the maximum likelihood estimator, which is NP-hard to find, can achieve small loss if beta is larger than the noise power, which is sqrt{ k log(k) }.
Since ML is impractical, the paper compares several computationally tractable methods. One method is based on matricization of the tensor, for which we require beta > k^{1/4} n^{k/4 - 1/2} to recover the vector with high probability. The other method is Power Iteration, which requires beta > k^2, but it needs the initial estimate used in the iteration to be approximately aligned with the true vector v0. The paper introduces Hybrid Power Iteration, which is a Power Iteration algorithm that starts from an initial estimate obtained by using the singular values of the matricized tensor. The final approach is based on Approximate Message Passing (AMP) using additional side information (in the form of noisy measurement of v0).
The paper has some numerical experiments comparing these methods.

This is a good paper. The theoretical results of this paper seem to be rigorous. It analyzes both what is achievable information theoretically, and also studies some computationally feasible methods. There is a large gap between the lower bound and upper bounds unless one uses side information in the high-dimensional limit of AMP.
The paper is written clearly, but it has many typos.

* Lemma 2.1: How concentrated is lambda_1(Z) around its expectation?

* The results of Section 4 are different from the previous sections as they are about the limit of n --> infty. Is this because of the "state evolution" technique? Is it possible to analyze the finite dimensional setting?

* Shouldn't C in Theorem 3 be larger than 1? Otherwise, the two regimes for beta would overlap.

* Typos:
- L065: extra "one"
- L091: omitted "information"
- L203: "fiund"
- L211: "typer"
- L220: "we" propose
- L251: extra 'e' after "the"
- L266: "wher"
- L302: "Bexause"
- L304: "thee"
- L351: "alforithm"
- L368: "imptrovement"

* L123 of the Supplementary Material: There is a factor of 2 and we have a factor of 24 at L116, so the RHS of Lemma 1.1 should have a factor of 48.
Summary: This is a good paper. The theoretical results of this paper seem to be rigorous, and the experiments validate the theory.
Author Feedback
Author rebuttal: Reviewer 43

- Tensor PCA for general tensors is hard to approximate, see
reference [14], Theorem 1.5 and Corollary 1.6.

- Of course, NP-hardness has no direct implications on the spiked model, but our analysis suggests that --for certain choices of the parameters-- this is also hard.
Partial support for this belief can be obtained by considering planted random XORSAT problem (suitably perturbed) . It is easy to see that this is directly connected to Tensor PCA, and work on planted random CSPs suggest that this is indeed hard to approximate.

- Two types of tractable approximation algorithms are used in the literature, that fall into power methods or matricization.

- The referee is correct in noting that the tensor structure on w imposes additional constraints. A more careful analysis shows that these constraints change the result only marginally.

- Our experience is that simulations for k=3 are fairly representative, since the major difference occurs between k=2 (matrix) and k>=3, but we agree that adding curves for k>3 can be informative.

%%%%%%%%%%%%%%%%%%%%%%%%%%%%%%%%%%%%%%

Reviewer 44

- \lambda_1(Z) concentrates subexponentially around its expectation: this can be proved by standard Gaussian isoperimetry.

- Yes, state evolution has only been established asymptotically. It should be possible to analyze some iterative algorithms in the finite-n case by using contraction-type arguments. We expect the resulting bounds to be looser than the state evolution results, but they would be interesting as they are non-asymptotic.

- We thank the referee for pointing out the typos, and with C > 1 in theorem 3. These will be corrected in the paper.